# Engineering bacterial vortex lattice via direct laser lithography

Daiki Nishiguchi [1,2,3], Igor S Aranson[4], Alexey Snezhko[5] & Andrey Sokolov [5]

A suspension of swimming bacteria is possibly the simplest realization of active matter, i.e. a class of systems transducing stored energy into mechanical motion. Collective swimming of hydrodynamically interacting bacteria resembles turbulent flow. This seemingly chaotic motion can be rectified by a geometrical confinement. Here we report on self-organization of a concentrated suspension of motile bacteria *Bacillus subtilis* constrained by two-dimensional (2D) periodic arrays of microscopic vertical pillars. We show that bacteria self-organize into a lattice of hydrodynamically bound vortices with a long-range antiferromagnetic order controlled by the pillars' spacing. The patterns attain their highest stability and nearly perfect order for the pillar spacing comparable with an intrinsic vortex size of an unconstrained bacterial turbulence. We demonstrate that the emergent antiferromagnetic order can be further manipulated and turned into a ferromagnetic state by introducing chiral pillars. This strategy can be used to control a wide class of active 2D systems.

[1] Pathogenesis of Vascular Infections Unit, Institut Pasteur, 75015 Paris, France. [2] Service de Physique de l'Etat Condensé, CEA, CNRS, Université Paris-Saclay, CEA-Saclay, 91191 Gif-sur-Yvette, France. [3] Department of Physics, The University of Tokyo, Hongo 7-3-1, Tokyo 113-0033, Japan. [4] Department of Biomedical Engineering, Pennsylvania State University, University Park, PA 16802, USA. [5] Materials Science Division, Argonne National Laboratory, Argonne, IL 60439, USA. Correspondence and requests for materials should be addressed to I.S.A. (email: isa12@psu.edu) or to A.S. (email: sokolov@anl.gov)

Hydrodynamic turbulence is associated with high Reynolds numbers ($Re \geq 5 \times 10^3$) and domination of inertia over viscous forces. Swimming microorganisms, such as common motile bacteria *Bacillus subtilis* or *Escherichia coli*, live in an exceedingly low Reynolds number world ($Re \sim 10^{-5} - 10^{-4}$). In contrast with active systems of crawling or dividing bacteria[1,2], cytoskeletal extracts[3,4], motile mammalian cell cultures[5,6], and self-propelled colloids[7–10], the collective motion emerges at relatively dilute bacterial concentrations, of the order of 3–5% of the volume fraction. The volume fraction typically measures the relative volume of bacterial bodies. While the volume of long flagella is negligible compared to the body volume due to their tiny diameter $\simeq 20$ nm, rotating flagella increase the effective volume fraction due to active stress distributed over the body and flagella. Bacterial "turbulent" swimming patterns are manifested by recurring vortices and jets with the length scales and velocities significantly exceeding the sizes and swimming speeds of individual bacteria[11–16]. However, the underlying physics of bacterial turbulence[17–19], the statistical properties[13,20,21], and the energy spectra of the self-organized bacterial turbulence[16,22] are fundamentally different from that of conventional (Kolmogorov) fluid turbulence.

Although unconstrained bacterial turbulence in the bulk has been described by a variety of continuum theories[16,19,22–24] and reproduced by discrete numerical models[16,25–27], it remains elusive how a geometrical confinement or obstacles modify the macroscopic behavior of the bacterial turbulence even in 2D. Swimming bacteria confined inside a small 2D circular chamber self-organize in a single-stable vortex[28,29]. It has been shown that hydrodynamic coupling between chambers with bacterial vortices can lead to the emergence of self-organized vortex lattices with short-range ferromagnetic or antiferromagnetic orders[30,31]. Here, we show that bacteria swimming around the bottom surface of a pendant drop self-organize in a lattice with anti-ferromagnetic order in the presence of periodic arrays of tiny microscopic pillars. While the area fraction occupied by pillars is only ~5%, we reproducibly observed the emergence of stable vortex arrays with a long-range order if the period of pillar array, $a$, is in the range from 60 to 90 μm. We demonstrate that the direction of bacterial vortices and the order of the vortex lattice can be controlled by chiral pillars.

## Results

### Self-organization of swimming bacteria into a vortex lattice.
While conventional turbulent flows can be suppressed by periodic structures, honeycombs or square grids[32,33], such periodic structures may reorganize non-steady bacterial motion into a periodic lattice of vortices. In this work we investigated swimming dynamics of bacteria *Bacillus subtilis* and the emergence of horizontal lattices of bacterial vortices in the presence of periodic arrays of tall and thin vertical pillars, see Methods section for details. In contrast to the studies by Wioland et al.[30] who investigated bacterial vortex arrays in hydrodynamically coupled microfluidic chambers, the total area (and volume) fraction of pillars in our work is much smaller. Thus, we impose significantly smaller geometric constraints on the bacterial turbulence (Fig. 1d and Supplementary Movie 1). Furthermore, in our experiments a bacterial suspension is exposed to air, which not only increases bacterial motility but also eliminates a solid–liquid interface unavoidable with microfluidic chamber experiments. We also simultaniously observe the dynamics of turbulent bacterial motion thanks to the existence of unconstrained area without pillars around the lattices, which is absent in the microfluidic chamber experiments[28,30,31]. Due to a principally different experimental set-up, the emerged bacterial vortex lattices demonstrate only antiferromagnetic order, while a system of connected chambers[30] exhibits the transition to a short-range ferromagnetic order with the increase of gap sizes. We explain this apparent discrepancy by a different mechanism of interactions between the vortices in these two systems. Since the pillars are only 14-μm wide, the bacteria are not able to swim along the liquid–solid boundaries of the pillars and to self-organize in a stable circulating loop around each pillar. Correspondingly, the dynamics of this system cannot be described by dual interacting vortex lattices, between and around pillars, as in ref. [30]. Instead, an array of tiny pillars creates a periodic set of stationary points with zero bacterial velocities. The emerged pattern arises from the continuity of bacterial flow between the pillars, as shown in Fig. 1c, e.

To examine the role of lattice geometry, we performed similar experiments on hexagonal lattices of hexagonal pillars (Fig. 1b and Supplementary Movie 6). We tested the lattice constants $a = 40$ μm and $a = 45$ μm to make the size of a unit cell comparable with the stable vortex size $\approx 70$ μm extracted from the square lattice experiments. Our experiments reveal that the spin orientations in a hexagonal lattice are random. These observations are in agreement with those reported in ref. [30] in spite of different pillars geometry (see the Supplementary Notes 1 and 2 for details).

### Spatial and temporal properties of emerged lattices.
The emerged dynamical patterns are characterized by the absolute values of mean vorticity $\langle |\langle \mathrm{rot}\mathbf{v}(\mathbf{r}, t)\rangle_t| \rangle_{\mathbf{r} \in \mathrm{ROI}_a}$ and the enstrophy $\left\langle \langle [\mathrm{rot}\mathbf{v}(\mathbf{r}, t)]^2 \rangle_t \right\rangle_{\mathbf{r} \in \mathrm{ROI}_a}$ of the bacterial velocity field calculated over the region of interest (ROI). Here, $\mathrm{ROI}_a$ denotes a set of ROIs with the lattice constant $a$. For small lattice constants, $a < 60$ μm, the bacteria are not able to develop turbulent motion, and their collective swimming is suppressed by densely placed pillars. As the lattice constant $a$ increases, both the mean vorticity and the mean enstrophy increase, reaching maximum at $a = 60$–$90$ μm (Fig. 2a). For these lattice periods, stable antiferromagnetic vortex lattices were observed. These lattice periods are comparable with doubled correlation length or the flow scale observed for the unconstrained (unbounded) bacterial suspension[13,14].

Stability of the bacterial vortex is characterized by temporal fluctuations of the full and tangential component of the bacterial velocity in the vortex $\sigma_{\mathrm{f,t}}(a) = \left\langle \sqrt{\left\langle \left[ \mathbf{v}_{\mathrm{f,t}}(\mathbf{r}, t) - \langle \mathbf{v}_{\mathrm{f,t}}(\mathbf{r}, t)\rangle_t \right]^2 \right\rangle_t} \right\rangle_{\mathbf{r} \in \mathrm{ROI}_a}$. The calculated values normalized by the root mean square (rms) velocities $v_{\mathrm{f,t}}^{\mathrm{rms}}(a) = \left\langle \sqrt{\left\langle [\mathbf{v}_{\mathrm{f,t}}(\mathbf{r}, t)]^2 \right\rangle_t} \right\rangle_{\mathbf{r} \in \mathrm{ROI}_a}$ are shown in Fig. 2b as a function of the pillar lattice constant $a$. Both $\sigma_{\mathrm{f}}/v_{\mathrm{f}}^{\mathrm{rms}}$ and $\sigma_{\mathrm{t}}/v_{\mathrm{t}}^{\mathrm{rms}}$ exhibit minima at $a = 70$ μm. At this lattice constant value, the vortices are hydrodynamically stabilized and the bacterial suspension self-organizes in a most stable coherent vortex lattice. The vorticity field $\langle \mathrm{rot}\mathbf{v}(\mathbf{r},t)\rangle_t$ calculated from the PIV clearly demonstrates the antiferromagnetic order inside the pillar arrays for the lattice spacing between 60 and 90 μm (Fig. 1a). For larger periods, $a > 100$ μm, the bacterial suspension is quasi-turbulent due to the reduced influence of the pillars. The mean vorticity is reduced as well due to large temporal fluctuations, while the mean enstrophy remains almost constant.

The antiferromagnetic order of vortices can be quantified with a spin–spin correlation, similar to that introduced in ref. [30]. A

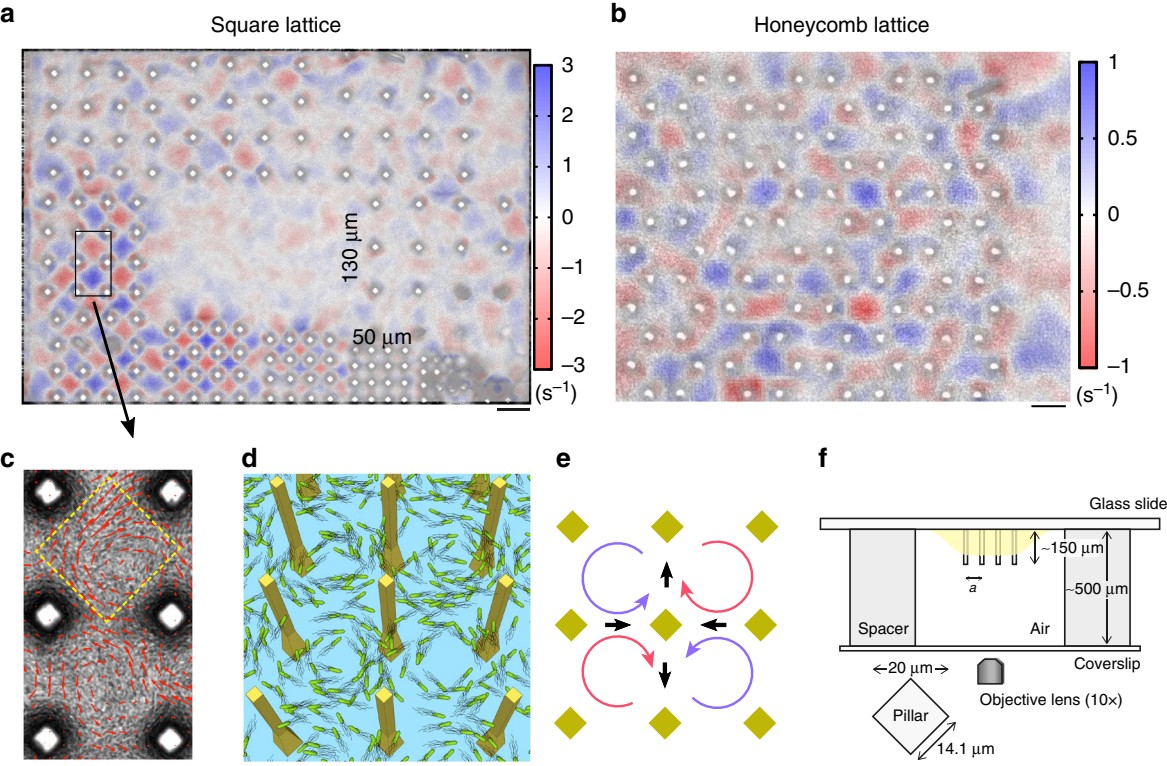

**Fig. 1** Self-organized bacterial vortex lattices. **a** A snapshot of bacteria swimming between square latices of pillars overlaid with a color plot of the average vorticity magnitude $\langle \text{rot}\mathbf{v}(\mathbf{r}, t)\rangle_t$. Pillars are arranged in nine arrays with different lattice constant $a$ increasing from 50 μm to 130 μm (clockwise) in 10 μm increment. The 40-μm lattice is excluded from analysis due to damaged pillars. Scale bar: 100 μm. **b** Distribution of the average vorticity magnitude for a honeycomb lattice. Scale bar: 50 μm. **c** Close-up of the rectangular area shown in **a**. Arrows indicate instantaneous velocities. Yellow dashed line depicts a single ROI area. Scale bar: 50 μm. **d** An artistic representation of bacteria swimming between 3D-printed micropillars (yellow). **e** Swimming bacteria self-organized in a lattice of vortices with antiferromagnetic order due to hydrodynamic interaction between adjacent vortices. Black arrows indicate the direction of bacterial flow between vortices. **f** A schematics of the experimental setup. For clarity, only one set of pillars is shown

spin variable for each ROI in the lattice at time $t$ is defined as

$$S_{i,a}(t) := \frac{\hat{\mathbf{z}} \cdot \left[ \sum_{\mathbf{r} \in \text{ROI}_a^i} (\mathbf{r} - \mathbf{r}_i) \times \mathbf{v}(\mathbf{r}, t) \right]}{\sum_{\mathbf{r} \in \text{ROI}_a^i} |\mathbf{r} - \mathbf{r}_i|}, \quad (1)$$

where $\text{ROI}_a^i$ denotes the $i$-th ROI in the lattice with period $a$, $\mathbf{r}_i$ is the geometrical center of $\text{ROI}_a^i$, and $\hat{\mathbf{z}}$ is the unit vector in the vertical direction. The magnitude of the spin represents the relative strength of a vortex and the sign of the spin reflects the predominant direction of rotation, positive for counterclockwise and negative for clockwise. Signs of the adjacent spins are the same for ferromagnetic order and alternate for antiferromagnetic order. To introduce the order parameter, we calculated the adjacent spin correlation $\chi_a(t)$ for each lattice constant $a$,

$$\chi_a(t) := \frac{\sum_{i \sim j} S_{i,a}(t) S_{j,a}(t)}{\sum_{i \sim j} \left| S_{i,a}(t) S_{j,a}(t) \right|}, \quad (2)$$

where the sum $\sum_{i \sim j}$ runs over all adjacent pairs in a single lattice structure. For a small lattice period of $a = 50$ μm, the order parameter fluctuates near −0.25 and occasionally drops to ≈−0.75. Densely placed pillars decrease average bacterial swimming velocity and prevent self-organization into a stable lattice. The observed antiferromagnetic order for $a = 60$–90 μm is characterized by strong anti-correlation between the adjacent vortices, $\chi_a \approx -1$ (Fig. 2e). For $a > 100$ μm, at the quasi-turbulent regime, the average order parameter $\langle \chi_a \rangle_t$ is relatively small, but temporal fluctuations of $\chi_a(t)$ are large (Fig. 2c, e and Supplementary Movie 2). For a short

period of time, the swimming bacteria self-organize into a large-scale coherent structure with a characteristic scale of the order of lattice period. However, the large vortices quickly break down into smaller vortices, with a more favorable scale of ~60–80 μm. Importantly, we did not observe any self-organization into a stable lattice for $a = 130$ μm, which is roughly the doubled period of the most stable lattice constant $a = 70$ μm and intrinsic length scale of vortices. That emphasizes a high sensitivity of the bacterial vortex pattern to defects in pillar arrays.

Previous experiments demonstrated that bacteria swimming in a microfluidic channel exhibits a sharp transition from a stable directed flow to a turbulent state with the increase in the channel width over ≈70 μm[34]. We observed the corresponding transition at a similar scale in a system with a noticeably smaller degree of confinement: the volume/area fraction of pillars is ~4 % for $a = 70$ μm. This observation highlights the importance of this characteristic scale for both fully enclosed and slightly confined systems and suggests a new less-invasive methodology to control and rectify the bacterial behavior at microscopic scale.

We investigated temporal properties of the observed patterns. Since the period of the antiferromagnetic vortex lattice is equal to the doubled period of the pillar lattice, the stable anti-ferromagnetic lattice has two possible spatial configurations. One configuration transforms to another by shifting along the main axes by $a$. Although such a transition requires simultaneous sign flipping of all spins in a lattice and is hardly observable in experiments, occasionally, a single bacterial vortex may flip the spin (Fig. 2d). An important question here is how the vortex size

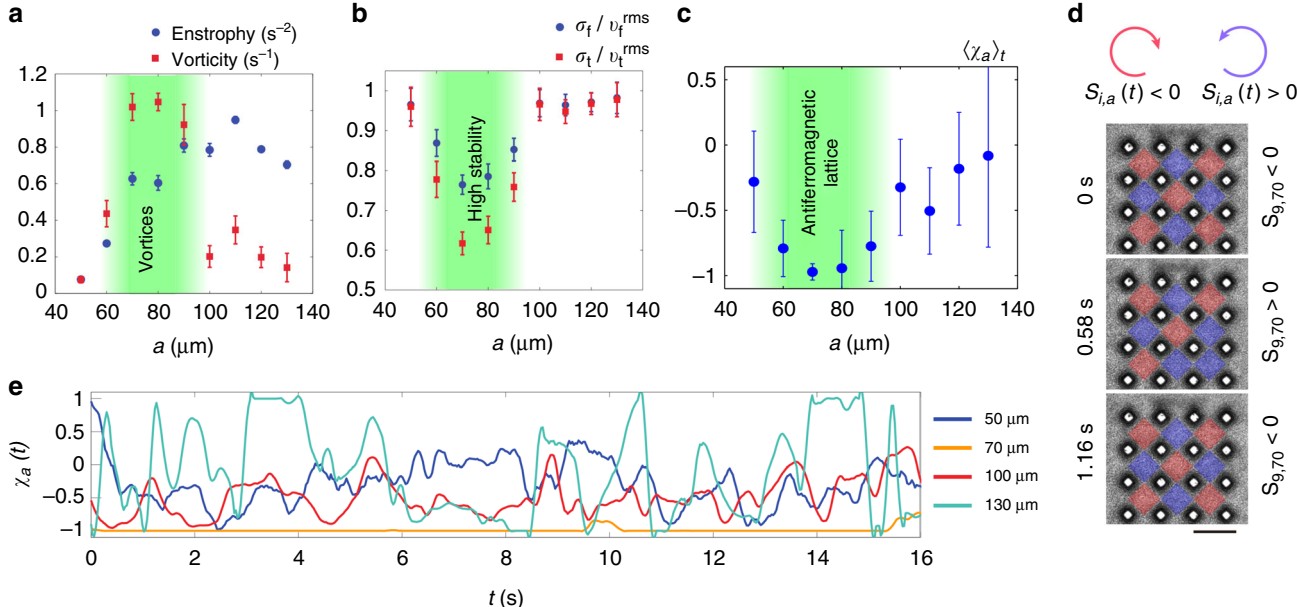

**Fig. 2** Spacial and temporal properties of emerged vortex lattices. **a** Mean enstrophy $\left\langle\left\langle[\mathbf{rot v}(\mathbf{r},t)]^2\right\rangle_t\right\rangle_{\mathbf{r}\in\mathrm{ROI}_a}$ (blue circles) and absolute values of mean vorticity $\left\langle|\langle\mathbf{rot v}(\mathbf{r},t)\rangle_t|\right\rangle_{\mathbf{r}\in\mathrm{ROI}_a}$ (red squares) vs lattice constant $a$. Error bars are estimated from the standard deviations among ROIs with the same $a$ (see the Supplementary Note 1). **b** Dependence of temporal fluctuations of full velocity $\sigma_f/v_f^{\mathrm{RMS}}$ (blue circles) and tangent velocity $\sigma_t/v_t^{\mathrm{RMS}}$ (red squares) on the lattice constant $a$. Both fluctuations have minima at 70 μm, at which the lattice exhibits stable antiferromagnetic order. Error bars are estimated from the standard deviations among ROIs with the same $a$. **c** Time average values of adjacent spin correlations $\langle\chi_a(t)\rangle_t$ vs the lattice constant $a$. Strong antiferromagnetic order $(\langle\chi_a(t)\rangle_t \simeq -1)$ is observed around $a \simeq 70\,\mu$m. Error bars: standard deviations of time series of $\chi_a(t)$. **d** Scale bar: 100 μm. Instantaneous distributions of signs of spins for a lattice with $a = 70\,\mu$m. A spin in the corner spontaneously changes the direction for a short period of time. **e** Examples of temporal dynamics of spins $\chi_a(t)$ for different lattice constants $a$ represented by different colors

and the spin–spin interaction in a lattice of different period affect the stability of a single vortex. To answer this question, we measured the mean persistence (or life) time of vortices in arrays with different periods. Each vortex in a pillar array is bound with four adjacent vortices, while a vortex on the edge interacts with a turbulent bacterial bath and therefore is less stable. To minimize the influence of such fluctuations, we fabricated larger arrays of pillars ($9 \times 9$ instead of $4 \times 4$) for periods $a = 50$–$90\,\mu$m (Fig. 3a and Supplementary Movies 3, 4, 5).

By analyzing the spins dynamics, we obtained the probability $P_a(t)$ of a spin to remain oriented in the same direction for a time $t$. This probability drops quickly for $a = 60\,\mu$m and $a = 90\,\mu$m, (Fig. 3b) due to chaotic behavior of the bacterial flow. The exponential decay of $P_a$ is a manifestation of Poisson random process: the probability of switching remains the same for any given period of time. The spin persistence time $\tau_a$ was estimated by fitting the experimental data with $P_a(t) \propto \exp(-t/\tau_a)$. Existence of stable antiferromagnetic lattice for $a = 60$–$80\,\mu$m facilitates the coherent spin dynamics. Hydrodynamic interaction between vortices increases the probability of each spin to remain in the preferable antiferromagnetic state and decreases the probability to remain in ferromagnetic state (Fig. 3c). Correspondingly, we observed two intrinsically different persistence times for this range of lattice constants: $\tau_a^{\mathrm{short}}$ and $\tau_a^{\mathrm{long}}$. In stable antiferromagnetic configuration for $a = 70\,\mu$m, each spin retains its favorable local antiferromagnetic orientation for $\tau_a^{\mathrm{long}} \approx 40\,$s, while very rarely flipping to unfavorable local ferromagnetic orientation for $\tau_a^{\mathrm{short}} \approx 0.2\,$s (Fig. 2d). This time $\tau_a^{\mathrm{short}}$ is significantly shorter than a typical time scale of bacterial dynamics ~1 s. The magnitude of the spin fluctuates around zero for quasi-turbulent lattices, $a \geq 100\,\mu$m.

Interaction of swimming bacteria inside the same ROI can be quantified by the spatial correlation function $C_a(r)$ and the correlation length $L_a$ of the velocity field $\mathbf{v}(\mathbf{r},t)$:

$$C_a(r) := \frac{\left\langle\left\langle\langle\mathbf{v}(\mathbf{r}',t)\cdot\mathbf{v}(\mathbf{r}'+\mathbf{r},t)\rangle_{\mathbf{r}',\mathbf{r}'+\mathbf{r}\in\mathrm{ROI}_a^i}\right\rangle_i\right\rangle_t}{\left\langle\left\langle\langle|\mathbf{v}(\mathbf{r}',t)|^2\rangle_{\mathbf{r}'\in\mathrm{ROI}_a^i}\right\rangle_i\right\rangle_t}. \qquad (3)$$

As expected, the correlation function $C_a(r)$ decays with $r$ (Fig. 3d). The presence of a vortex inside each ROI is manifested by negative values of $C_a$ at large $r$, which is especially noticeable for $a = 70$–$90\,\mu$m. The increase of $C_a(r)$ for $r > 70\,\mu$m is observed in the arrays with large periods $a > 100\,\mu$m. This is a hint to the coexistence of two vortices in the same ROI, which leads to frustration and destabilization of the antiferromagnetic order. The correlation length $L_a$ is defined as the distance at which $C_a(r)$ becomes smaller than $1/e$. $L_a$ increases with $a$ as expected and approaches the correlation length of unconstrained suspension $L_\infty \simeq 45\,\mu$m (Fig. 3d).

**Self-organization of swimming bacteria around chiral pillars.** So far the studies were limited to non-chiral pillar arrays. Chirality is an important factor controlling the organization of collective bacterial motion. However, fabrication of chiral obstacles with controlled shapes used to be prohibitively difficult. Thanks to a recent progress in a two-photon photolithography, we 3D-printed arrays of hollow chiral towers, 30 μm in diameter and 100 μm in height. The tip of each tower has holes oriented at 45° (positive towers) or 135° (negative towers) to the radius of the tower (Fig. 5a). Bacteria, swimming from and to the center of each tower through these holes, create a vortex (Fig. 5b). The spin state of the vortex is prescribed by the "chirality" of the tower. Depending on the pre-manufactured order of a chiral tower array

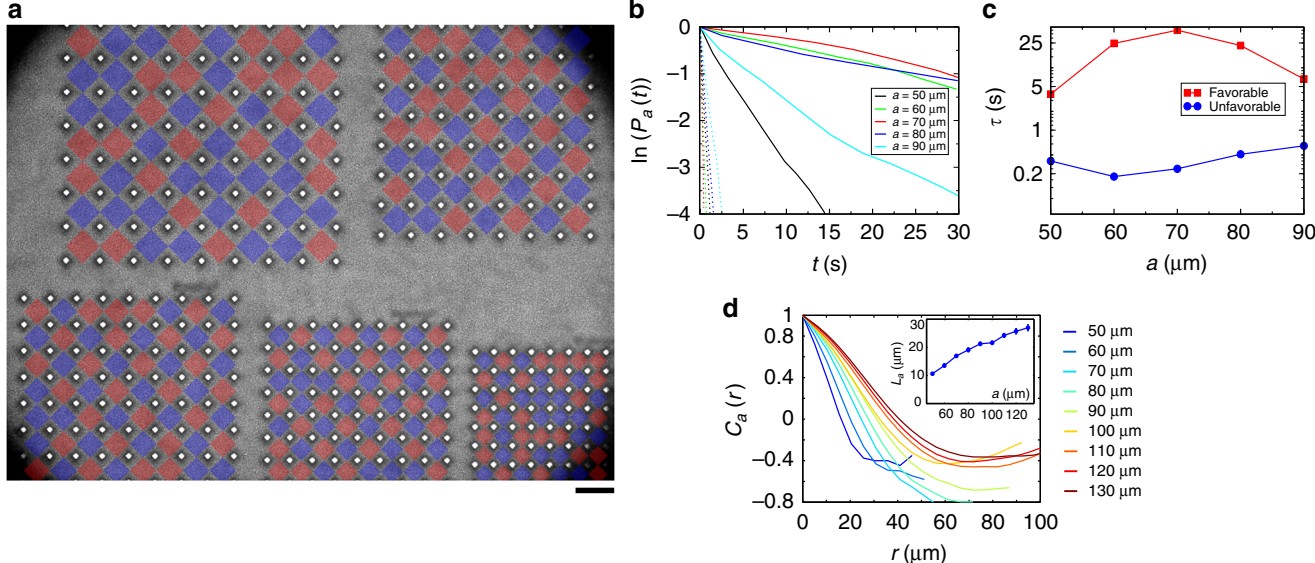

**Fig. 3** Stability of antiferromagnetic order and velocity correlation functions. **a** Large arrays (9×9) of pillars for $a = 50–90\,\mu m$ with a step of 10 μm. A perimeter of vortex lattices (8×8) is excluded from further image processing, leaving the internal region of the size 4×4 for analysis. Scale bar is 100 μm. **b** Semi-log plots of the vortex persistence probability $P_a(t)$ for different lattice constants $a$. Solid and dashed lines correspond to probabilities of a vortex to remain in antiferromagnetic and ferromagnetic orientation correspondingly. **c** Semi-log plots of persistence times of a vortex (see the Supplementary Note 1 for details) as a function of lattice constant. For $a = 60–80\,\mu m$, stable antiferromagnetic order significantly increases the persistence time $\tau_a^{long}$ for the preferred antiferromagnetic vortex state (red squares) and decrease the time $\tau_a^{short}$ for more unfavorable local ferromagnetic orientation (blue circles). **d** Correlation functions of the velocity field $\mathbf{v}(\mathbf{r},t)$ inside the lattice structures with different lattice constants. Inset: Correlation lengths as a function of the lattice constant $a$. The correlation length in unconstrained suspension is $L_\infty \simeq 45\,\mu m$

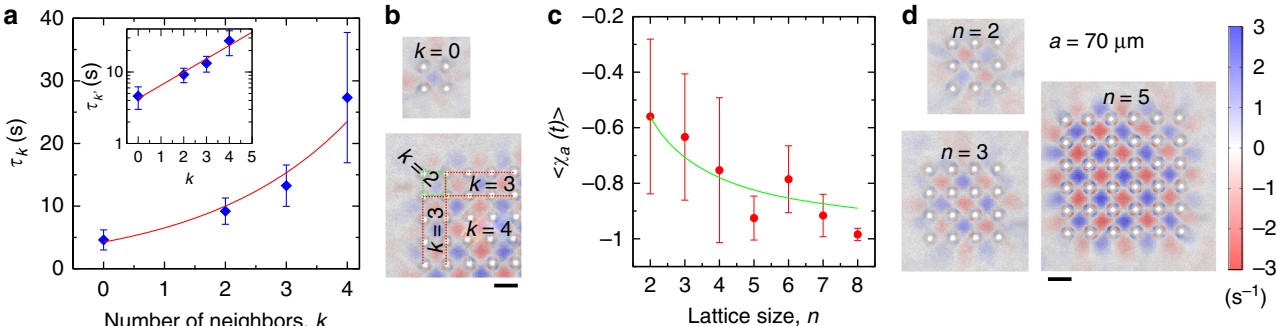

**Fig. 4** Persistence time and order parameter dependencies. **a** Persistence time $\tau_k$ of a vortex with different number of neighbors $k$, shown in **b**. Symbols depict experimental data, solid line is a fit to theoretical dependence $\tau_k \sim \tau_0 \exp(\alpha k)$, with $\tau_0 = 4.25$ s, $\alpha = 0.43$, see Eq. (9). Inset: $\tau_k$ vs $k$ in semi-log scale. Error bars: standard errors of measurements. **b** Location of bacterial vortices with different number of neighbors $k$. Scale bar: 70 μm. Color bar is the same as that in **d**. **c** Order parameter $\langle \chi_a(t) \rangle_t$ for a whole vortex lattice (including perimeter) of the size $n \times n$. Lattice constant $a = 70$ μm. Symbols shows experimental data, solid line is a fit $\langle \chi_a(t) \rangle_t = -1 + 0.88/n$. Error bars are calculated from the standard deviations of time series $\chi_a(t)$. **d** Vorticity map of a bacterial flow for lattices of different sizes $n \times n$. Shown examples for $n = 2, 3, 5$. Lattice constant (distance between pillars) $a = 70$ μm, see the Supplementary Fig. 11 for other values of $n$. Scale bar: 70 μm

(antiferromagnetic or ferromagnetic), the bacterial suspension self-organizes into a stable antiferromagnetic state with a period of the tower lattice, $a$, or a "double-lattice" state according to definition in ref. [30] (Fig. 5c). The second state is represented by two ferromagnetic lattices of opposite spins shifted by a half-period along both crystallographic axes. By combining positive, negative, and neutral (not chiral) towers in different patterns, we can produce various types of vortex lattices and control their stability.

## Discussion

Fluctuations suppress a long-living antiferromagnetic order in relatively small arrays of pillars. The stability of any finite size lattice clearly depends on a fraction of vortices located on its perimeter, exposed to a turbulent bacterial bath. In contrast to the microfluidic chamber experiments with completely confined bacterial turbulence[28,30,31], the coexistence of the destabilizing bulk turbulence and the stable vortex lattices in our experiments enable us to assess such effects in detail. First, we calculated the persistence time $\tau$ as a function of nearest neighbors number $k$ (Fig. 4a and Supplementary Movie 7–10). Depending on the position in the lattice, $k$ assumes a value of 2 on the vertices, 3 on the edges, and 4 in the bulk (Fig. 4b). We also printed an array of 4 pillars (just a single vortex) to attain the value of $k = 0$. The dependence of $\tau$ vs $k$ shows an exponential increase, consistent with our theoretical prediction based on the Kramers escape rate

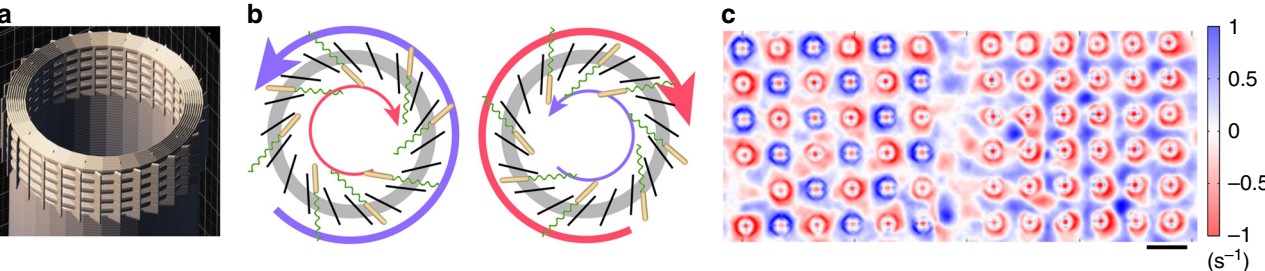

**Fig. 5** Bacterial vortex lattices in the presence of chiral towers. **a** 3D model of chiral tower. The top part consists of five 3-μm thick hollow disks spaced at 3-μm vertical distance from each other. These disks are supported by 1-μm thin vertical walls oriented at 45° relative to the radius. **b** Chiral towers create internal and external hydrodynamical bacterial vortices due to specially oriented holes in their tips. **c** Vorticity map of bacterial flow between antiferromagnetic (left) and ferromagnetic (right) chiral tower arrays. Scale bar: 100 μm

(Methods section). We measured the order parameter for the whole lattice, including perimeter, for different array sizes from $n \times n = 2 \times 2$ to $8 \times 8$, while keeping the lattice constant $a = 70$ μm (see examples in Fig. 4d). The results are shown in Fig. 4c. Since the peripheral vortices are more exposed to turbulent bath fluctuations, in small arrays of $n = 2$–4, unstable vortices on the perimeter fluctuate frequently and destabilize the entire lattice. For larger arrays of $n = 5$–7, their influence is reduced and becomes negligible for $n \geq 8$. The dependence of the order parameter vs $n$ is consistent with the $1/n$ law. The behavior can be inferred from the fact that the ratio of less coherent peripheral vortices to the bulk vortices scales as $1/n$ (Supplementary Notes 3 and 4).

Our study provides an insight into self-organization of concentrated bacterial suspension under seemingly insignificant geometrical constraints. Indeed, a periodic array of tiny pillars, taking only 3–5% of total suspension volume, drastically alters chaotic bacterial swimming pattern and turns it into a periodic vortex array. The emerged square vortex lattice reveals several surprising observations. First, in contrast with previous microfluidic experiments[30], the apparent antiferromagnetic order at large distances between pillars suggests a different mechanism for the bacterial vortices interactions. The concept of the double-lattice model[30] cannot be applied due to the small pillar sizes. Antiferromagnetic order arises from direct hydrodynamic interaction between the adjacent vortices. Second, the observed patterns demonstrated significantly higher persistence and much larger magnitude of the order parameter in spite of small volume/area of pillars: Adjacent spin–spin correlation achieves the value close to −1 for $a = 70$ μm. This high vortex lattice robustness prevents penetration of defects from the perimeter of the lattice to the bulk. For the lattices with the $a = 70$ μm, the spins can only flip near the border of the lattice, while the lattice remains antiferromagnetic in the bulk. Our work provides novel strategies for minimally invasive control of active matter that may be applicable to other experimental systems exhibiting vortex formation under geometrical confinement such as active colloids[35,36], cytoskeletal extracts[3,4], and vibrated grains[37]. Self-organization of bacteria in nearly perfect vortex lattices can be used as a tool for more efficient energy extraction by an array of gears driven by swimming bacteria[38,39] or control of turbulent active motion of bacteria in Newtonian[11,14] or anisotropic fluids[40,41].

## Methods

**Pillars manufacturing**. The pillars consist of a photopolymer resist (IP-Dip) and are 3D-printed on a glass slide by direct laser lithography[42], Photonic Professional GT system, Nanoscribe GmbH. Being 150-μm tall and 14-μm wide, the pillars are arranged in square lattices of the period $a$ ranging from 50 μm to 130 μm with 10 μm increment (Fig. 1a). The central part of the experimental cell is left pillar-free so that parameters of the unconstrained bacterial motion can be monitored

simultaneously. In addition, we made honeycomb lattices with lattice constants $a = 40$ μm and $a = 45$ μm.

**Bacteria preparation**. The bacteria *Bacillus subtilis* (strain 1085) were inoculated on an LB agar plate (Sigma-Aldrich) and stored in a refrigerator at temperature 4 °C. A night before the experiment a small amount of bacteria from the agar plate was transferred to liquid growth medium Terrific Broth (TB) and incubated at 30 °C overnight. Next morning bacteria were washed, concentrated by centrifugation, and placed in fresh TB medium at the final concentration of $10^{10}$ cm$^{-3}$. The average length of bacteria is 5–8 μm (not including 10–15-μm-long flagella with a diameter of $\simeq 20$ nm) with a diameter of 0.8 μm. Correspondingly, the final volume fraction of bacteria is $\approx$ 3–4%. At this packing fraction the bacteria swim collectively with a typical speed $\approx$ 50–60 μm s$^{-1}$ [13].

**Experimental procedure**. A droplet of a concentrated suspension was placed on a glass slide with the printed arrays of microscopic pillars. The thickness of the droplet was slightly smaller than the pillar's height to ensure that the pillars pierce the droplet. The droplet was enclosed by a plastic spacer and by a coverslip with an air gap of about 0.5 mm. The enclosure minimized evaporation of water from the droplet while providing oxygen to bacteria. Subsequently, the experimental cell was flipped so that the bacteria accumulate at the bottom of the droplet due to the gravity and aerotaxis (Fig. 1f). The dynamics of bacteria were captured by an Olympus IX71 inverted microscope and a high-resolution (5120 × 3840) HS20000C camera at ×10 magnification at 30.0–52.7 fps depending on the experiments.

**Image processing**. The velocity field $\mathbf{v}(\mathbf{r}, t)$ of bacterial motion was calculated by custom particle image velocimetry (PIV) MATLAB scripts (Fig. 1b). The PIV subwindows were 20 × 20 μm and separated by every 5 μm (75% overlap), the spatial resolution being smaller than any characteristic scales of the observed collective motion. Estimation of the bacterial velocity in the proximity of a pillar is challenging due to several factors: A meniscus creates an optical distortion in the vicinity of each pillars complicating bacteria tracking; the imposed geometrical confinements on bacterial motion near the pillars lead to a significant vertical motion and reduces the accuracy of spatial tracking. To avoid this problem, we excluded areas around each pillar from our analysis and measured the bacterial swimming parameters only in square regions of interest (ROI) between pillars (Fig. 1b). We used only the red pixels of this RGB color camera for analysis because the images acquired by the red pixels (longer wavelength) had the highest spatial resolution due to a smaller amount of scattering and diffraction of transmitted light.

**Analytical estimate of the persistence time**. To estimate the persistence time as a function of the number of neighbors, we use a coarse-grained approach where individual spins are described by an angle variable $\phi$ with the corresponding spin value $V = \cos(\phi)$. The angle $\phi$ is governed by the following equation

$$\frac{d\phi}{dt} = -\gamma \sin(2\phi) + \xi(t), \tag{4}$$

where $\gamma$ is the relaxation rate, and $\xi(t)$ is a Gaussian white noise approximating interaction with the turbulent bacterial bath, $\langle \xi(t) \rangle = 0$, $\langle \xi(t)\xi(0) \rangle = 2D\delta(t)$, $D$ is the noise intensity. Obviously, the choice of the sin function in Eq. (4) is not unique. Qualitatively similar results can be obtained for arbitrary symmetric bi-stable function.

In the absence of noise, a solution to Eq. (4) relaxes to either 0 or $\pi$. Correspondingly, the spin variable tends to $V = \pm 1$, representing clockwise/counterclockwise rotating vortices (compare to the approach in ref. [30]). With the noise, the system switches between these symmetric steady states. The persistence time $\tau_0$ of an isolated spin can be estimated from the Kramers escape rate[43] (since

the system needs to overcome the energy barrier $\gamma$ to switch from the state 0 to $\pi$ and vise verse)

$$\tau_0 \sim \exp(\gamma/D) \qquad (5)$$

Interacting spins $\phi_{ij}$ on square lattice can be described similarly,

$$\frac{d\phi_{ij}}{dt} = -\frac{dH}{d\phi_{ij}} + \xi_{ij}(t) \qquad (6)$$

where the free energy functional $H$ is of the form

$$H = -\frac{1}{2}\gamma \sum_{ij} \cos(2\phi_{ij}) + \eta \sum_{ij} \sum_{lm} \cos(\phi_{ij} - \phi_{lm}). \qquad (7)$$

Here $\eta > 0$ is the interaction parameter, and the sum $\sum_{lm}$ is taken with respect to all nearest neighboring sites on a square lattice.

In the absence of noise, Eq. (6) favors a stable antiferromagnetic lattice corresponding to the minimum of the free energy $H$. With fluctuations, the spins flip, and the persistence time depends on the number of nearest neighbors $k$. One can find an upper bound for the energy barrier $E$ vs $k$ ($k = 0$ for an isolated spin, $k = 2$ for vertices, $k = 3$ for edges, and $k = 4$ inside the lattice):

$$E \approx \gamma + 2\eta k \qquad (8)$$

It gives the following estimate for the persistence time $\tau_k$

$$\tau_k \sim \tau_0 \exp(2\eta k/D) \qquad (9)$$

The exponential dependence for the persistence time $\tau_k$ given by Eq. (9) is in excellent agreement with the experiment (Fig. 4a).

## Data availability

The data in support of the reported findings and computer code are available from the corresponding author upon request (sokolov@anl.gov).

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

## Acknowledgements

The research of A.Sokolov, I.S.A., and A.Snezhko (experiments and data analysis) was supported by the US Department of Energy, Office of Basic Energy Sciences, Division of Materials Science and Engineering. D.N. (experiments, image processing, and data analysis) was supported by the Advanced Leading Graduate Course for Photon Science (ALPS) and a Grant-in-Aid for Japan Society for Promotion of Science (JSPS) Fellows (Grant No. 26-9915).

## Author contributions

A.Sokolov conceived the study. D.N. and A.Sokolov performed experiments, D.N. carried out the image processing and data analysis, I.S.A. contributed with theoretical analysis. D.N., I.S.A., A.Snezhko, and A.Sokolov analyzed and interpreted the experimental results. All authors contributed in writing the manuscript.

## Additional information

**Competing interests:** The authors declare no competing interests.

