## [Peer Review File · Nature Communications]

Reviewers' comments:

Reviewer #2 (Remarks to the Author):

This is a very nice paper that demonstrates control of collective bacterial dynamics using a new lithography technique, which allows to go beyond recent experimental studies as those in Ref [27] (Wioland et al, Nature Physics). In particular, the control of collective bacterial dynamics by chiral boundaries is fundamentally new. I much enjoyed reading the paper and think that it will be broad general interest.

The study has been carefully performed and the experiments and analysis are of high quality.

I have however a suggestion regarding some parts of the presentation/discussion, which in my opinion should be improved further prior to publication. At the end of the first paragraph on page 2, the authors write

"These observations are in disagreement with those reported in Ref. [27] in spite of effectively large gaps (see the Supplemental Material for details [30])."

There exist a number of fundamental differences between the experimental setup reported in [27] and that in the present paper:

First, the setup in the present paper appears to have a liquid-air interface, whereas that in [27] has only solid-liquid interfaces.

Second, the experiments in [27] used triangular pillar geometry for hexagonal lattices whereas the present paper uses rectangular pillar geometries.

Third, according to the fifth SI movie and Fig. S7, there appear to be significant turbulent flows present in the pillar-free boundary domain that can feed back into the rather small pillar-domains and lead to the favoring of disordered states. The experiments in [27] did not have that pillar free boundary domains.

Fourth, as mentioned by the authors themselves, the pillar diameter appear to be quite different. In fact it seems to me that the ratio of pillar size and gap size in the hexagonal experiments of the present paper is quite small that the pillars effectively do not matter anymore. This does in fact *agree* with the observations reported in [27], see Fig. 3c,d in [27], which show that ferromagnetic order disappears for gap sizes larger than $\sim 20\text{-}25\text{ }\mu\text{m}$. According to the scale bar in Fig. S7, the gap size in the present experiments also seems to be larger $\sim 20\text{-}25\text{ }\mu\text{m}$, so the results appear to consistent with those in Ref. [27].

In view of the substantial differences in setups, it is generally not that surprising that the authors observe somewhat different types of order regimes. However, as explained above, I think that there is actually less disagreement for the hexagonal lattices than the authors' statement in the main text suggests.

I therefore recommend that the authors reformulate the corresponding text parts, by more carefully discussing the differences in the respective setups.

Reviewer #4 (Remarks to the Author):

See attached file.

Reviewer #5 (Remarks to the Author):

The authors report on the self-organisation of bacteria into vortex arrays, when collectively swimming in patterned environments. Using microfabricated pillars, and quantitative PIV measurements, Nishiguchi et al clearly establish that weak geometrical constrains are sufficient to stabilize dynamical patterns with a high level of antiferromagnetic order. The paper is closed showing that the directions of the individual vortices can be effectively controlled from the microscopic chirality of the pillars.

The control over the bacterial dynamics is very impressive, and the quality of the data as well. However I feel that there exists a gap between the claims summarised in the introductory paragraph and the main text and the actual experimental results. I believe that these results should ultimately be published in a high-impact journal, however, I do not feel that this manuscript warrants publication in Nat. Comm in the present form.

Either more quantitative experiments should be done to support some of the central claims, or the main conclusions should be clarified.

Main concerns:

Firstly, The authors state in the introductory paragraph that they "show that bacteria self-organize into a lattice of hydrodynamically bound vortices with a long-range antiferromagnetic order". This conclusion is not supported by the two sets of experiments performed in lattices as small as 4×4 and 9×9 pillars. A systematic study of the decay of the order parameter with the system size is required to clearly establish the existence of long-range order. This task is usually extremely difficult to achieve in 2D. To the best of my knowledge, this type of results has been very rarely demonstrated in active-matter physics, and one noticeable exception comes from the authors, ref [20].

Secondly, I feel that the presentation of the results should also be more accessible to nonspecialists. I am not familiar with this type of experiments, and I could not find basic informations necessary to appreciate some of the conclusions of the article. e.g. what is the size of a cell? What is its swimming speed? What is the packing fraction of the suspension? Mentioning a density of 10^{10} cm^{-3} is not really helpful to readers unfamiliar with this type of systems. The experimental setup would also deserve to be more clearly introduced in the main text.

Minor comments:

-Why did the authors resort to a two-point function as an order parameter? Why not using a simple, and standard, staggered magnetization?

- I am not sure that any conclusion can be easily drawn from Fig. 2e. The fluctuation amplitude looks very similar for all the experiments.

Summary of main changes

1. We modified Figure 1 and added new Figure 4.
2. We included the results of our new additional experiments on a study of the decay of the order parameter with the system size.
3. We included new theoretical model and analysis on the dependence of the persistence time on the number of neighbors.
4. We processed experimental data to extract persistence times as a function of number of neighbors. This data is used to demonstrate the validity of the suggested theoretical model.
5. We clarified the differences and similarities between the setup and results in our work and Ref.27 (Wioland, et al. Nature Physics 2017).
6. We modified the main text following the recommendation of reviewers. The main modifications in the main text is highlighted by red texts.
7. We added in SI new sections “III. Lattice size scaling experiment” and “IV. Theoretical description: exponential behavior of persistence time”.
8. In accordance with new analysis and new experiments, we added 6 new supplemental movies (5, 7-10) and renumbered them.
9. Several reference papers were added to answer criticisms and questions raised by the Reviewers.

Reviewer #2 (Remarks to the Author):

This is a very nice paper that demonstrates control of collective bacterial dynamics using a new lithography technique, which allows to go beyond recent experimental studies as those in Ref [27] (Wioland et al, Nature Physics). In particular, the control of collective bacterial dynamics by chiral boundaries is fundamentally new. I much enjoyed reading the paper and think that it will be broad general interest.

We sincerely thank the Reviewer 2 for spending his/her time on our paper, very positive judgment and the valuable comments and suggestions.

The study has been carefully performed and the experiments and analysis are of high quality.

I have however a suggestion regarding some parts of the presentation/discussion, which in my opinion should be improved further prior to publication. At the end of the first paragraph on page 2, the authors write

“These observations are in disagreement with those reported in Ref. [27] in spite of effectively large gaps (see the Supplemental Material for details [30]).”

There exist a number of fundamental differences between the experimental setup reported in [27] and that in the present paper:

First, the setup in the present paper appears to have a liquid-air interface, whereas that in [27] has only solid-liquid interfaces.

It is indeed a significant difference and we included such notice in the main text so that readers unfamiliar to this topic can easily recognize the differences between our experiment and that in [27]. Thank you for pointing it out.

Second, the experiments in [27] used triangular pillar geometry for hexagonal lattices whereas the present paper uses rectangular pillar geometries.

It is true that the shapes of the pillars are different in these two experiments. However, in fact, we printed hexagonal pillars instead of square pillars for hexagonal lattice experiments to avoid unwanted obstruction of vortical flow in the lattices by the edges of the pillars. Although the shapes are different in our experiment and that in [27], this basic idea of determining the shapes of the pillars is common and we believe that this slight difference do not alter our results qualitatively. Nevertheless, we appreciate the Reviewer for pointing out this important point because we forgot to mention that the shapes of the pillars are hexagonal for the hexagonal lattices in the old manuscript. We added remarks both in the main text and in SI.

Third, according to the fifth SI movie and Fig. S7, there appear to be significant

turbulent flows present in the pillar-free boundary domain that can feed back into the rather small pillar-domains and lead to the favoring of disordered states. The experiments in [27] did not have that pillar free boundary domains.

This is one of the main focus of our experiments in comparison with the previous experiments done in confined microfluidic devices as in [27]. In the microfluidic chambers, turbulent bacterial suspensions were forced to form vortices even in a single independent circular chamber. What is demonstrated in [27] is that the coupling between “*preformed*” vortices in each chamber can lead to ferromagnetic or antiferromagnetic lattices. In contrast, bacterial turbulence is *not* confined in our experiments, and we demonstrated that such bulk unconstrained bacterial turbulence, when it encounters the periodic arrays of pillars, can self-organize in stable antiferromagnetic vortex lattices even in the presence of destabilizing turbulent flows around the pillar arrays. In this sense, unlike [27], we extracted order out of chaotic motion. Therefore, having the pillar-free boundary domains is crucial in our experiments. Furthermore, having the pillar-free area allows us to estimate the swimming speed and characteristic length of turbulent motion, and improve the control of the experiment.

To clarify the advantage and the implications of our experiments with pillar-free turbulent areas in comparison with [27], we performed the additional study both experimentally and theoretically on how this area may affect the dynamics of a small lattice by changing the lattice sizes. Indeed, if the lattice is small, it is less stable due to exposure to a turbulent bacterial bath. For a large lattice ($n > 7$) this influence is negligible.

Fourth, as mentioned by the authors themselves, the pillar diameter appear to be quite different. In fact it seems to me that the ratio of pillar size and gap size in the hexagonal experiments of the present paper is quite small that the pillars effectively do not matter anymore. This does in fact **agree** with the observations reported in [27], see Fig. 3c,d in [27], which show that ferromagnetic order disappears for gap sizes larger than ~ 20 -25 μm . According to the scale bar in Fig. S7, the gap size in the present experiments also seems to be larger ~ 20 -25 μm , so the results appear to consistent with those in Ref. [27].

We thank the Reviewer for the important remark. Indeed, for hexagonal lattices our observations are in agreement with [27]. The text is modified.

In view of the substantial differences in setups, it is generally not that surprising that the authors observe somewhat different types of order regimes. However, as explained above, I think that there is actually less disagreement for the hexagonal lattices than the authors’ statement in the main text suggests.

We agree that there is not so much disagreement especially in the hexagonal lattices since our experiments setup is different. However, as we explained above in this reply, there is a crucial difference in that we observe antiferromagnetic vortex lattice formation for the square lattices, a different order obtained in [27], even in

the presence of unconstrained turbulent areas that destabilizes the vortex order in the lattices. We clarified the differences in the setup and the significances of our results in the main text, and added additional analysis related to this.

I therefore recommend that the authors reformulate the corresponding text parts, by more carefully discussing the differences in the respective setups.

Again, we would like to sincerely thank the Reviewer for his/her detailed suggestions and comments that are useful for improving our manuscript and scientific understandings.

Reviewers #4

We thank the Reviewer for very careful reading our manuscript and constructive criticisms. We significantly modified the manuscript to address his/her concerns and suggestions.

This manuscript reports a study of the effect of a widely-spaced lattice of pillars, non-chiral or chiral, on the emergence of collective motion in a bacterial suspension in a density regime where, without the pillars, the cells show active turbulence. It was found that even a sparse array of non-chiral pillars organised the bacterial turbulence into an 'antiferromagnetic' lattice, while chiral pillars gave rise to ferromagnetic ordering.

The antiferromagnetic ordering in the case of non-chiral pillars is the focus of attention. This observation was deemed 'surprising' because it differed from previous work [27]. In this previous work, confined bacterial vortices were allowed to interact through a gap between the vortices, and a transition from antiferromagnetic to ferromagnetic ordering was observed when the gap sized increased. The authors of the current work claim that their observation of antiferromagnetic ordering using widely-spaced non-chiral pillars 'in spite of effectively large gaps' is surprising. In other words, the naïve reader might map the previous results for large gap size to the present geometry, and expect ferromagnetic ordering, but in fact antiferromagnetic ordering is found. The observation of no ordering of any kind in the current work for hexagonal lattices is again contrasted with previous work, where ferromagnetic ordering was found for a hexagonal array of vortices.

We clarified the differences between our experimental setup and [27]. We also rephrase a few statements to avoid an impression that certain findings in [27] may be inaccurate or that our results are in contradiction with [27].

Given that this is not the first study of interacting bacterial vortices, the authors seem to base their claim to novelty on the observed differences with [27]. But how surprising is it that the finds here differ from those reported in [27]? As the authors themselves remind us, albeit only in the Supplementary Information, the previous work has already 'highlight[ed] the importance of slight difference in boundary conditions for understanding macroscopic behavior, especially the emergent order.' So, by changing completely how bacterial vortices interact with each other, it is hardly surprising that the observations will change dramatically, as is found. So, what the current manuscript effectively does is to emphasize that the authors of [27] were completely correct – change the boundary conditions, and everything changes.

The boundary conditions are indeed very important. A crucial difference between our experiment and [27] is the degree of confinement, or the existence of unconstrained bulk turbulent area: In the microfluidic chambers as in [27],

turbulent bacterial suspensions were forced to form vortices due to boundaries even in a single independent circular chamber. What is demonstrated in [27] is that the coupling between “*preformed*” vortices in each chamber can lead to ferromagnetic or antiferromagnetic lattices. In contrast, bacterial turbulence is *not* confined in our experiments, and we demonstrated that such bulk unconstrained bacterial turbulence, when it encounters the periodic arrays of pillars, can self-organize in stable antiferromagnetic vortex lattices even in the presence of destabilizing turbulent flows around the pillar arrays. In this sense, unlike [27], we extracted order out of chaotic motion.

We highlighted this difference and the interaction mechanism in our new manuscript, and in order to further clarify the advantage and the implications of our experiments with pillar-free turbulent areas in comparison with [27], we performed the additional study both experimentally and theoretically on how this area may affect the dynamics of a small lattice by changing the lattice sizes. This advances our understanding on dynamics of fluctuations and order in bacterial turbulence.

Moreover, in [27] I find a simple theoretical model, which was shown to fit the data presented, while the present work presents purely observations supported only by a brief qualitative explanation (see my comments further below on the brevity of explanation). It would have been useful to know, for example, whether the model proposed in [27] can be adapted to discuss the current experiments, and whether doing so would give any new insights.

As the model proposed in [27] assumes circulating currents around the pillars that do not exist in our experimental setup due to tiny sizes of the pillars, it is impossible to adapt this model to our experiment. Removing such circulating currents from this model leads to a trivial uninteresting model. Nonetheless, in the revised manuscript, we developed a new theoretical model in a different approach. Based on the Kramers escape rate theory, we evaluated the dependence of persistence time vs number of neighbors on the lattice. The theory predicts an exponential increase of the persistence time. Our experimental data from previous and new additional experiments validate this model.

There is also a rather large claim at the end of the first paragraph, viz., that the way vortex arrays is manipulated here ‘can be applied to other active 2D systems’. If at least one, preferably non-bacterial, experimental example is actually given in this work, then the claim may be justified. As it is, the large claim is just that, a large claim!

To give clear ideas of experimental systems possibly controllable by our approach, we identify several examples of non-bacterial experimental systems that form vortices in confinement: active colloidal rollers, cytoskeletal extracts, and vibrated grains. Of course, it cannot be completely clear whether these systems are actually controllable with our approach and it needs to be demonstrated experimentally, but this is apparently beyond the scope of this current manuscript.

In fact, we are currently in a process of preparation of another paper on the experiments performed with artificial active particles (non-bacterial). While our results can not be fully transferred to this system, we observed some similarities. In this sense, we believe our approach is quite promising.

In more detail, I have the following comments.

Introduction: the bacterial volume fraction quoted in the second paragraph for the emergence of turbulence, 3-5%, is presumably the volume fraction of cell bodies, without taking any account of the (long!) flagella. This should be made clear.

Flagella do not contribute so much to actual volume fraction. The diameter of the flagella is about 20 nm and that of the cell body is 0.8 μm (~40 times larger). Even if we consider that the flagella are twice to three times longer than cell bodies, the volumes of the cell bodies are at least 500 times larger than those of flagella. However, it is true that rapidly rotating flagella might effectively occupy larger volumes in suspension. We added several sentences clarifying these parameters and the effective volumes so that unfamiliar readers can easily assess our experimental systems. We thank the Reviewer for pointing out this confusing point.

In the next paragraph, I presume there has been no convincing observation of bacterial turbulence in 3D. Again, this should be made clear.

We corrected this point to make it clear that all the previous observations on vortex self-organization in microfluidic chambers are done in 2D. We appreciate this Reviewer's careful comment on references.

Results: A crucial experimental detail is missing. Nowhere are we told that the lattice parameter is incremented in steps of 10 μm – this information is, as far as I can see, buried in second paragraph of section IA in the supplement.

Although we had already mentioned the increment of 10 μm in the first submitted manuscript in the caption of Fig.3 and in the 7th line of "Pillar manufacturing" subsection of Materials and Methods Section in the main text, it is true that readers who start reading from the beginning cannot find this information easily when they see Fig.1 and 2 at first. We added this information to Figure 1 caption.

Right at the end of this paragraph, we find in one sentence (with a reference to one image and one schematic) the sum total of the author's discussion of the coupling between vortices in this work. Given that the very basis of their claim to novelty is a different coupling mechanism from [27], I think a somewhat fuller discussion, accompanied by figures for comparable quality to the very clear Figure 1 in [27], is called for.

We added the Figure 1e, which illustrates the continuity of bacterial flow and clarify the mechanism of coupling between vortices. Our new phenomenological model is also based on this concept and is now supporting our claim.

Also, it would be useful to know whether the observations depend on the height of the sample cell, especially since no one knows really how bacterial turbulence is affected by confinement in the vertical direction (see my earlier remarks about the lack of observations of truly 3D turbulence).

We agree that investigation of the height effect would be interesting and of great importance. However, unfortunately the experimentally achievable height of pillars is limited by two factors. To observed dynamics of bacteria on the surface, the vertical thickness of a drop has to be smaller than the pillars heights. First, the 3D-printed pillars should not be too long to avoid gravitational depression of the bottom surface. As can be seen in our Fig.1(a) and Fig.S7, these very thin (high aspect ratio) 3D-printed pillars are quite fragile and easily detached or distorted. Therefore, saying that two-photon direct laser lithography technology enables printing high-aspect-ratio structures, achieving significantly high-aspect-ratio structures is still quite challenging and experimentally time-consuming. Second, the drop should be relatively thick to minimize the effect of evaporation and relative shrinking of the drop. Because of these two limiting factors, we choose the pillar heights of 150 μm . As the Reviewer him/herself mentioned, 3D structure of bacterial turbulence has not yet sufficiently investigated, so focusing on the effect of the depth of suspensions is indeed interesting and should be focused in future.

In the final sentence of the results section, the authors claim that ‘by combining positive, negative and neutral (not chiral) towers in different patterns, we can produce various types of vortex lattices and control their stability.’ This is a rather general claim. I see no evidence for it, beyond the two examples shown in Fig. 4c, which I’m afraid is not adequate for justifying the considerably larger claim with which the authors end this paragraph. (I find no further justification of this claim in the Supplement either.)

We demonstrated the possibility to create ferromagnetic and antiferromagnetic lattices by manipulating the chirality of towers. Furthermore, we can create desirable antiferromagnetic order with chiral towers at our will, whereas with neutral towers emergent antiferromagnetic order has only two possible configurations appearing according to initial conditions. By designing arrays with different chirality we can create various patterns in which positive or negative vortices are placed wherever we want. However, for the moment, we still do not have clear idea on the designs of patterns useful or meaningful for micro-engineering as we mentioned at the end of the Conclusion section, and so we did not play with other patterns but rather focus on these two basic patterns. Further investigations will indeed provide more details.

In the Discussion, the authors claim that their findings ‘provides new experimental insights into bacterial turbulence’. I find it hard to discern what new insights are provided into bacterial turbulence as such, since the finds are about regular lattices;

neither do I understand what would constitute a new experimental insight. Do the authors mean insight into how to do experiments? Or what? Perhaps the authors mean that their work provided new experimental insight into how to control bacterial turbulence. In any case, I suggest a rewording of this claim.

We thank Reviewer for the suggestion. Indeed, we provided the insight into control and self-organization of active suspensions. This is the first sentence in the paragraph. We remove the claim regarding “insight into bacterial turbulence”.

Overall, while this work is indeed of considerable interest to a specialist active matter physics audience, it does not seem to justify publication in a journal such as Nature Communications in this form because at least some of the main claims to novelty and general importance appear weak. The lack of any theoretical framework for interpreting the observation also means that the manuscript as it stands falls short of what one might expect from a submission to this journal.

Our work combines novel experimental and theoretical results and is of interest to very broad scientific communities, from biological physics to engineers developing potentially new microfluidic devices. We believe that revised manuscript fits Nature Communications.

Again, we would like to sincerely thank the Reviewer for his/her detailed suggestions and comments based on his/her careful reading and profound knowledge that are useful for improving our manuscript.

Reviewer #5 (Remarks to the Author):

The authors report on the self-organisation of bacteria into vortex arrays, when collectively swimming in patterned environments. Using microfabricated pillars, and quantitative PIV measurements, Nishiguchi et al clearly establish that weak geometrical constrains are sufficient to stabilize dynamical patterns with a high level of antiferromagnetic order. The paper is closed showing that the directions of the individual vortices can be effectively controlled from the microscopic chirality of the pillars.

We sincerely thank the Reviewer for careful reading of our manuscript and critical comments. Here are the replies to the his/her critique in order.

The control over the bacterial dynamics is very impressive, and the quality of the data as well. However I feel that there exists a gap between the claims summarised in the introductory paragraph and the main text and the actual experimental results. I believe that these results should ultimately be published in a high-impact journal, however, I do not feel that this manuscript warrants publication in Nat. Comm in the present form.

There are several claims made in the introductory paragraph:

- 1) Self-organization into a lattice of hydrodynamically bounded vortices with the antiferromagnetic order. This is the main claim of the paper and supported by technical analysis, several pictures (Fig1a, Fig 2d, Fig 3a, Fig 5c) as well as measurements of the order parameter.
- 2) The stability and almost perfect antiferromagnetic order for lattice constant $a=60-80\mu\text{m}$. This claim is also supported by our analysis and Fig2b,c, Fig 3b,c.
- 3) The order may be manipulated by introducing chirality into the system. According to our measurements of vorticity (Fig5c), the order indeed can be controlled by chirality. However, we corrected the text to avoid misunderstanding.

We also added the theoretical analysis to eliminate the gap between claims and observations. Using the Kramers escape rate theory, we derived how the persistence time depends on the number of neighbors with respect to the square lattices. The predicted exponential dependence is in good agreement with experiments. We also predicted how the order parameter depends on the lattice size.

Either more quantitative experiments should be done to support some of the central claims, or the main conclusions should be clarified.

We performed additional experiments on the dependence of the persistence time on the number of neighbors and the dependence of the order parameter on the lattice

size. These experiments confirm our theoretical prediction on the persistence time dependence.

Main concerns:

Firstly, The authors state in the introductory paragraph that they "show that bacteria self-organize into a lattice of hydrodynamically bound vortices with a long-range antiferromagnetic order". This conclusion is not supported by the two sets of experiments performed in lattices as small as 4x4 and 9x9 pillars. A systematic study of the decay of the order parameter with the system size is required to clearly establish the existence of long-range order. This task is usually extremely difficult to achieve in 2D. To the best of my knowledge, this type of results has been very rarely demonstrated in active-matter physics, and one noticeable exception comes from the authors, ref [20].

We thank the Reviewer for this important remark. Of course, we are aware of this point. However, it is almost impossible to extract the lattice size dependence of the order parameter for several decades and to see how it converges to -1 or how the fluctuations are suppressed (exponentially or algebraically) due to the limited field of view. In ref [20] by one of the Authors, such analysis was possible because the order can be calculated by continuously increasing the area used for calculation, but in the case of this current work, if we need to assess it precisely, we need to simultaneously observe pillar lattices with the sizes of, for example, 1x1, 10x10, 100x100, and so on, which easily exceeds our experimentally accessible field of view. We can, of course, decrease the magnification of the objective lens to see such large pillar arrays, but this leads to unreliable data. These are why we weren't able to do such finite size scaling analysis at first.

Nonetheless, to make our statement as quantitative as possible, we performed new experiments and investigated the effect of the system size in detail. Because large lattices such as 10x10 or 100x100 cannot be observed simultaneously, we constructed pillar arrays of 2x2 to 8x8. As shown in the new Fig. 4, the order parameter is decreasing from approximately -0.5 to -1 with the size of the lattice from 2x2 to 8x8. From this experimental observation and the decay of order parameter toward -1 consistent with 1/n law (new Fig/4c), we concluded again that, for large arrays of pillars, the influence of fluctuations on the bulk vortices are negligible and thus the emergent order can be regarded as long-ranged.

Secondly, I feel that the presentation of the results should also be more accessible to nonspecialists. I am not familiar with this type of experiments, and I could not find basic informations necessary to appreciate some of the conclusions of the article. e.g. what is the size of a cell? What is its swimming speed? What is the packing fraction of the suspension? Mentioning a density of 10^{10} cm^{-3} is not really helpful to readers unfamiliar with this type of systems.

We thank the Reviewer for pointing out that our current description is not reader-friendly. We added some important information to the main text so that unfamiliar readers can have a clear idea of the setup and properly assess our work.

The experimental setup would also deserve to be more clearly introduced in the main text.

We added explanations on the experimental setup in the introductory part of the main text, such as explicitly mentioning the difference between our setup and that in [27] (absence of a solid-liquid interface, existence of bulk unconstrained turbulence, shape of pillars, etc.).

Minor comments:

-Why did the authors resort to a two-point function as an order parameter? Why not using a simple, and standard, staggered magnetization?

We chose to characterize the order parameter similar to work by Wioland [27] to avoid misunderstanding when comparing our results with previous conclusions. Furthermore, we do not think that staggered magnetization is intuitively understandable by nonspecialist readers in such an interdisciplinary journal.

- I am not sure that any conclusion can be easily drawn from Fig. 2e. The fluctuation amplitude looks very similar for all the experiments.

Perhaps the Reviewer did not notice a yellow line at the bottom, always staying around -1. This figure demonstrates the typical amplitude and time scale of order parameter fluctuations for different a .

The key conclusion is following: For $a=50$ μm and $a=100$ μm , fluctuations are medium. For $a=70$ μm , fluctuations are negligible. For $a=130$, the system is chaotic.

Again, we would like to sincerely thank the Reviewer for his/her useful suggestions and comments that encouraged us to design new experiments for improving our manuscript.

REVIEWERS' COMMENTS:

Reviewer #2 (Remarks to the Author):

The authors have carefully addressed all my comments and questions in their revised version. The new experimental data and theoretical analysis added during the revision process has substantially improved the manuscript. As I stated in my previous report, I think this is a groundbreaking piece of work that demonstrates a new technique for controlling collective bacterial dynamics and will likely become a standard reference in the field. I am therefore recommending this excellent paper very strongly for publication in Nature Communications.

Reviewer #4 (Remarks to the Author):

The authors have substantially clarified the similarities and difference between their work and Wioland et al. I am now satisfied that there are substantial differences, which were obscure in the original submission. They have performed new experiments in response partly to my comment and partly to the comments of Reviewer 5, and built a simple theoretical model to predict successfully the outcome of these experiments. Overall, these are substantial and welcome changes, which to my mind has rendered the manuscript acceptable for publication in Nature Comm.

Reviewer #5 (Remarks to the Author):

The authors really set out to improve their manuscript addressing all comments and questions raised by the referees, including mine.

The new experiments and measurements shown in new Fig. 4, and the related simple theoretical arguments added in the Method section, are very nice additions.

Making a recommendation about this paper is a difficult task. The authors are all experts in active matter physics, they very clearly and honestly introduce their results compared to the existing literature. The results are new and supported by measurements that are optimal given the limitation of the experimental setup (however, the minor comments reported below should be considered). The experiment that are extremely technical and only very few groups in the world could have conducted them.

At this stage, the question is not the paper is good or not (it is of course a very good paper), but whether it is good enough to warrant publication in Nature Communications. My honest answer is: I do not know. I would recommend this manuscript to journals like Soft Matter or PRE without an ounce of hesitation. I am more hesitant to recommend it to a journal like Nat. Comm, as I feel that the research is too specialized given the previous contributions of the Goldstein group, and the intrinsic limitations of the setup to finite size systems.

Additional comments:

— The variations of the persistence time of antiferromagnetic order with the system size is very clear (The data are indeed consistent with a Kramers picture, however, they do not provide an unambiguous demonstration of exponential variations. The dynamical range is not broad enough to make such a strong statement).

— The main new point made by the authors is that the fluctuations in the bacterial dynamics mostly originates from the sample boundaries, which rather confirm my initial question about the possibility to make any prediction about the actual bulk phases.

Dear Dr. Dubrovina,

REPLY TO REVIEWERS' COMMENTS:

Reviewer #2 (Remarks to the Author):

The authors have carefully addressed all my comments and questions in their revised version. The new experimental data and theoretical analysis added during the revision process has substantially improved the manuscript. As I stated in my previous report, I think this is a groundbreaking piece of work that demonstrates a new technique for controlling collective bacterial dynamics and will likely become a standard reference in the field. I am therefore recommending this excellent paper very strongly for publication in Nature Communications.

We thank the Reviewers #2 for spending time on reading our paper, valuable suggestions in the previous round and positive final review.

Reviewer #5 (Remarks to the Author):

The authors really set out to improve their manuscript addressing all comments and questions raised by the referees, including mine.

The new experiments and measurements shown in new Fig. 4, and the related simple theoretical arguments added in the Method section, are very nice additions. Making a recommendation about this paper is a difficult task. The authors are all experts in active matter physics, they very clearly and honestly introduce their results compared to the existing literature. The results are new and supported by measurements that are optimal given the limitation of the experimental setup (however, the minor comments reported below should be considered). The experiment that are extremely technical and only very few groups in the world could have conducted them.

At this stage, the question is not the paper is good or not (it is of course a very good paper), but whether it is good enough to warrant publication in Nature Communications. My honest answer is: I do not know. I would recommend this manuscript to journals like Soft Matter or PRE without an ounce of hesitation. I am more hesitant to recommend it to a journal like Nat. Comm, as I feel that the research is too specialized given the previous contributions of the Goldstein group, and the intrinsic limitations of the setup to finite size systems.

We thank the Reviewers #5 for careful and detailed analysis of our work and evaluation of our paper as a very good paper. We also thank for questions raised during the review, which resulted in important refinements and additional discussions included in the final version of the paper. We believe that revised

manuscript provides important insights into not only properties of active systems but also possible applications in microengineering and thus suitable for publication in Nature Communications which has broad readership including physicists and engineers.

Additional comments:

— The variations of the persistence time of antiferromagnetic order with the system size is very clear (The data are indeed consistent with a Kramers picture, however, they do not provide an unambiguous demonstration of exponential variations. The dynamical range is not broad enough to make such a strong statement).

The range is limited by possible numbers of neighbors for each vortex, which changes from 0 to 4 for a square lattice. We would not make a statement based only on experimental data. However, our theoretical analysis predicts the exponential dependence. Experimental data is consistent with our theoretical prediction, as we wrote in the manuscript.

— The main new point made by the authors is that the fluctuations in the bacterial dynamics mostly originate from the sample boundaries, which rather confirm my initial question about the possibility to make any prediction about the actual bulk phases.

We agree that the fluctuations mostly come from the boundaries, while the bulk is much more stable. That statement is reflected in our paper.

Sincerely,
The authors.